# *In silico* analyses identify sequence contamination thresholds for Nanopore-generated SARS-CoV-2 sequences

**Ayooluwa J. Bolaji**[1,2], **Ana T. Duggan**[1] *

**1** Public Health Agency of Canada, National Microbiology Laboratory, Winnipeg, Canada, **2** Cadham Provincial Laboratory, Winnipeg, Canada

* ana.duggan@phac-aspc.gc.ca

**Data Availability Statement:** The data used in this publication has been made publicly available on Zenodo, https://doi.org/10.5281/zenodo.8206455.

## Abstract

The SARS-CoV-2 pandemic has brought molecular biology and genomic sequencing into the public consciousness and lexicon. With an emphasis on rapid turnaround, genomic data informed both diagnostic and surveillance decisions for the current pandemic at a previously unheard-of scale. The surge in the submission of genomic data to publicly available databases proved essential as comparing different genome sequences offers a wealth of knowledge, including phylogenetic links, modes of transmission, rates of evolution, and the impact of mutations on infection and disease severity. However, the scale of the pandemic has meant that sequencing runs are rarely repeated due to limited sample material and/or the availability of sequencing resources, resulting in the upload of some imperfect runs to public repositories. As a result, it is crucial to investigate the data obtained from these imperfect runs to determine whether the results are reliable prior to depositing them in a public database. Numerous studies have identified a variety of sources of contamination in public next-generation sequencing (NGS) data as the number of NGS studies increases along with the diversity of sequencing technologies and procedures. For this study, we conducted an *in silico* experiment with known SARS-CoV-2 sequences produced from Oxford Nanopore Technologies sequencing to investigate the effect of contamination on lineage calls and single nucleotide variants (SNVs). A contamination threshold below which runs are expected to generate accurate lineage calls and maintain genome-relatedness and integrity was identified. Together, these findings provide a benchmark below which imperfect runs may be considered robust for reporting results to both stakeholders and public repositories and reduce the need for repeat or wasted runs.

## Author summary

Large-scale genomic comparisons provide a wealth of knowledge, including modes of transmission, rates of evolution, and the impact of mutations on infection, disease severity, and treatment effectiveness. As a result, the public release of genomic data has proven crucial to response of the SARS-CoV-2 pandemic. However, studies continue to show that

**Funding:** This study was funded/supported by the Public Health Agency of Canada and Genome Canada through the Applied Genomics Innovation for Laboratory Excellence (AGILE) program (formerly the Canadian Public Health Laboratory Network COVID-19 Genomics Program (CCGP)) and the Canadian COVID-19 Genome Network (CanCOGeN), respectively. The funders had no role in study design, data collection and analysis, decision to publish, or preparation of the manuscript.

**Competing interests:** The authors have declared that no competing interests exist.

some of the genomic data in public repositories are contaminated from a variety of sources. For instance, the rapid response to the SARS-CoV-2 pandemic prevented many sequencing runs from being repeated, resulting in the occasional upload of imperfect runs to public repositories. It is of note that when genomic data is contaminated, both scientific decisions/studies and public health measures may be compromised. To identify genome contamination threshold(s) for SARS-CoV-2 sequences generated by Nanopore sequencing, computational biology techniques were used to generate artificially subsampled and contaminated libraries. This is the first study of its kind and we hope that the results obtained provide a starting point to investigate and report contamination for groups producing and analyzing NGS data.

## Introduction

Genomics and whole genome sequencing of pathogens provide vital information for disease transmission, identification of novel outbreaks, and vaccine candidate selection [1]. Numerous investigations have shown that in the early days of the COVID-19 pandemic, results from genomic monitoring were not only equivalent to epidemiological contact tracing data, [1] but also capable of tracing previously unidentified linked transmissions [2]. It is noteworthy that public health decisions were guided by genomic investigations in some jurisdictions to stop the spread of SARS-CoV-2, including travel bans and stay-at-home orders [1–3]. Thus, rapid whole genome sequencing for SARS-CoV-2 is essential for public health intervention.

Since the SARS-CoV outbreak in 2002–2003, the importance of genomic information for addressing outbreaks brought on by pathogenic coronaviruses has grown. Indeed, progress regarding the studies of this virus shifted dramatically as the complete viral genome was sequenced [4]. However, due to the technology available and the lag in data sharing, it took about 3 months to complete the sequencing of the first complete genome of the SARS-CoV virus [5,6]. Complete genomes were generated in 2002–2003 by first propagating the virus in cell lines, extracting viral RNA from these cell lines, and using a Sanger sequencing approach to produce complete and partial genomes [7]. It is worth noting that advances in genomics have significantly improved sequencing methodologies and timelines in less than two decades, owing to the development of third generation NGS and long-read sequencing technologies. Thus, in late December 2019, the first whole genome sequences of the novel beta coronaviruses, now known as SARS-CoV-2, was obtained using metagenomics and NGS approaches—supplemented with PCR and Sanger sequencing [8–10] and made available online within days. The availability of the SARS-CoV-2 reference whole genome sequences facilitated the development of real-time PCR-based diagnostic assays that helped to understand the transmission patterns and epidemiology of the virus [11]. Both partial and whole genome sequences of SARS-CoV-2 genomes have been reported from around the world and used to monitor the global spread of the virus.

Prior to the 2019–2020 SARS-CoV-2 pandemic, there were approximately 1200 complete betacoronavirus genomes in GenBank. As of July 2023, there were over 15.8 million SARS-CoV-2 genomes available in the Global Initiative on Sharing Avian Influenza Data (GISAID) (https://www.gisaid.org) platform, reflecting a significant increase in the number of available genomes throughout the pandemic. These genomic sequences are generated on different next-generation sequencing (NGS) devices, namely Illumina, Ion Torrent, Oxford Nanopore, and PacBio SMRT platforms. While sequencing technologies have error rates of varying degrees [12,13] genome sequence contamination may also occur during sample preparation and

sample processing at both wet and dry lab steps of the workflow. Also, contamination in reference databases is more concerning than contamination in individual sequencing studies and, according to a few studies, human DNA contamination has been found in non-primate reference genomes [14,15]. GenBank has been reported to contain millions of contaminated sequences, and human contamination in bacterial reference genomes has resulted in thousands of false protein sequences [16]. Therefore, even if researchers properly decontaminated or controlled for contaminants, contamination in reference databases runs the risk of tainting the results of many genomic studies. Further, numerous studies have identified a variety of sources of contamination in public NGS databases and these studies have discovered widespread cross-contamination between samples as well as contamination in sequencing kits and laboratory reagents [16–19].

While NGS is widely used for the rapid detection and characterization of positive COVID-19 cases, one of the drawbacks is that NGS runs are rarely repeated for reasons including limited funds to repeat expensive library preparation reactions even when samples are multiplexed. This has meant that in some cases, the results of imperfect runs are uploaded to public repositories and used to drive public health decisions. Most studies, with few exceptions, do not clearly define the quality control metrics used to include or exclude genomic data from public repositories. Thus, contamination can seriously affect the results of genomic analyses of organisms and viruses leading to spurious alignments and incorrect downstream variant calls.

For this study, we conducted an *in silico* experiment using known SARS-CoV-2 genomes produced from Nanopore sequencing. We assessed the effect of contamination on lineage calls and single nucleotide variants (as a measure of assembly accuracy) using sequences from the same variants and sequences from different variants. The effect of sequencing depth on contamination detection was further investigated using three different bins of reads (12,500 reads, 25,000 reads, and 50,000 reads) as a proxy for sequencing depth. For each sequencing depth, 14 artificially subsampled and contaminated genomes were generated. These samples were generated by mixing clinical SARS-CoV-2 samples *in silico* at different proportions to represent low (1% to 9%) and high (10%, 20%, 30%, 40%, and 50%) contamination levels. Results obtained in this study should help establish internal quality controls and contamination thresholds for SARS-CoV-2 sequences that can be extrapolated from the amount of reads identified within negative controls. Our goal is to provide groups generating the sequencing data with benchmarks to guide their decisions of which runs are of sufficient quality for submission to public repositories and to offer researchers a standard by which results obtained from contaminated SARS-CoV-2 runs can be trusted for variant calling and other downstream reporting.

## Methods

### SARS-CoV-2 genome sequencing and generation of the **i**n *silico* contaminated libraries

Due to low quantities of viral genomic materials in clinical swab specimens, most SARS-CoV-2 genomes are generated from a tiled amplicon sequencing approach. In this study, amplicons were generated with the Freed primer scheme [20], using tiling PCR and prepared for Oxford Nanopore Technologies sequencing using the ONT Ligation Sequencing Kit (SQK-LSK109) as per the manufacturer's guidelines. The resulting reads were basecalled using the Guppy high accuracy model (5.0.7) with default settings. The average number of reads generated from 812 clinical SARS-CoV-2 samples (representing Alpha, Delta and Omicron lineages) sequenced on MinION and GridION devices were determined using NanoStat (https://github.com/wdecoster/nanostat). Since ONT does not run for a fixed number of cycles, there is no

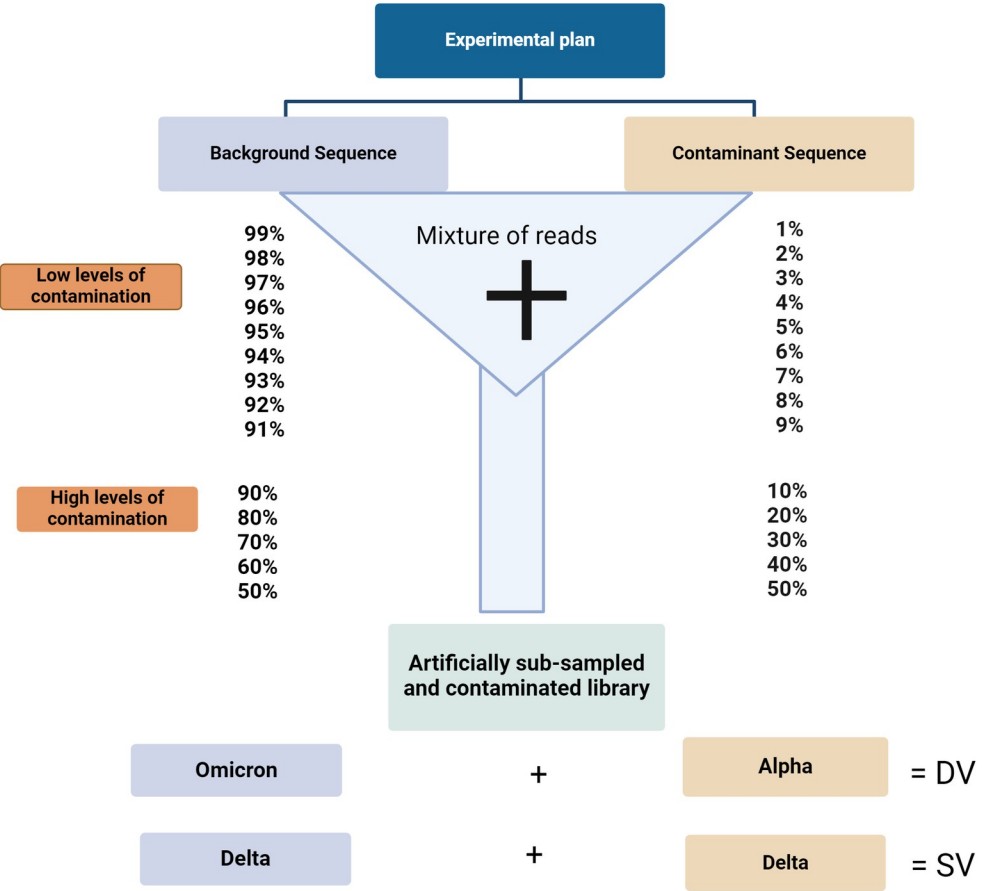

**Fig 1.** Experimental design of the artificially subsampled and contaminated genomes for the 14 levels of contamination (low and high levels) at three sequencing depths (low—12,500 reads, medium -25,000 reads, and high—50,000 reads). The controlled datasets were generated from known clinical SARS-CoV-2 samples. The sequence combinations were either known omicron sequences mixed with known alpha sequences or known delta sequences mixed with known delta sequences. Textual representation of experimental plan is depicted in Table 1. Created with BioRender.com.

chemistry-induced constraint to measure average run sizes. The results obtained were used as a guide for the selection of the read depths as well as experimental design for the generation of the artificial genomes, where low, medium, and high sequencing depths were represented by 12,500 reads, 25,000 reads, and 50,000 reads, respectively (Fig 1). Artificially subsampled and contaminated reads were generated with seqtk (https://github.com/lh3/seqtk) from 4 clinical sample libraries to represent the contributions of both the known background and contaminant genomes to the artificially contaminated libraries (Fig 1 and Table 1).

## Data processing

The artificially generated libraries were processed using a nextflow implementation of the ARTIC pipeline (https://github.com/connor-lab/ncov2019-artic-nf). Variant candidates were identified using Nanopolish (https://github.com/jts/nanopolish). Output files generated from the ARTIC pipeline were further processed using ncov-tools to perform quality control on sequencing results (https://github.com/jts/ncov-tools). Reads were mapped to the reference SARS-CoV-2 genome NCBI GenBank accession (MN908947.3) to further detect the major mutations and single nucleotide variants (SNVs) of consensus sequences for each sample.

**Table 1. Standardized terms and parameters of the artificially subsampled and contaminated genomes.**

| Artificially subsampled and contaminated genomes | Standardized term | Background genome | Contaminant genome | Sequencing depth |
|---|---|---|---|---|
| Low sequencing depth sample with contaminants from similar variant | LSD_SV | Delta—AY.25.1 genome | Delta–AY.27 genome | Low– 12,500 reads |
| Medium sequencing depth sample with contaminants from similar variant | MSD_SV | Delta—AY.25.1 genome | Delta–AY.27 genome | Medium– 25,000 reads |
| High sequencing depth sample with contaminant from similar variant | HSD_SV | Delta—AY.25.1 genome | Delta–AY.27 genome | High– 50,000 reads |
| Low sequencing depth sample with contaminants from different variant | LSD_DV | Omicron–BA. 1 genome | Alpha– B.1.1.7 genome | Low– 12,500 reads |
| Medium sequencing depth sample with contaminants from different variant | MSD_DV | Omicron–BA. 1 genome | Alpha– B.1.1.7 genome | Medium– 25,000 reads |
| High sequencing depth sample with contaminants from different variant | HSD_DV | Omicron–BA. 1 genome | Alpha– B.1.1.7 genome | High– 50,000 reads |

Number of consensus SNVs was determined by the number of variants found in the consensus file while the number of variants SNVs is the number of variants found in the iVar variants/ VCF files. Number of consensus 'N' was determined by the number of Ns (missing data) in the consensus sequences. Lineages were assigned using Pangolin (version 4.0.3: https://github.com/cov-lineages/pango-designation, pangoLEARN–version 1.2.333: https://github.com/cov-lineages/pangoLEARN). Scorpio call is the output of the constellation assigned to a query by Scorpio–a tool for classifying, haplotyping and defining Variants of Concern or Variants of Interest for a species. As all data used in this study originated from known clinical sample genomes, the metrics of the artificially contaminated libraries were compared against their "true" genome sequence.

The artificially generated datasets (raw reads) as well as their corresponding consensus sequences have been deposited to Zenodo: https://doi.org/10.5281/zenodo.8206455.

## Genome pairwise comparison and heat map

Aligned nucleotide consensus genome sequences of both the background and contaminant source clinical samples and the artificially generated genomes were imported to MEGA11 software to calculate pairwise distance. The observed p-distance option was chosen as input for the Model/Method setting while the default options were chosen for the other settings. The pairwise distances output table was imported as a text-delimited file into R v.4.1.1 and the ggplot2 v3.3.1 package was used to generate heat maps for data visualization.

## Results

### Global nucleotide comparison at different levels of contamination for different sequencing depths

By subsampling the sequences of a known clinical delta sample (AY.25.1) contaminated with reads from another known clinical delta sample (AY.27), we simulated 14 different scenarios to quantify the effect of contamination (Fig 1). To investigate the effect of both low and high levels of contamination on lineage calls and SNVs as a measure of assembly accuracy, we performed a series of global nucleotide comparisons using pairwise p-distance analyses. The distance (proportion) of variant nucleotide sites was compared and plotted as a heat map for all artificially generated samples at the three sequencing depths–low (12,500 reads), medium (25,000 reads), and high (50,000 reads) (Fig 2). These results show that, regardless of sequencing depth and contamination type (i.e., similar (Fig 2A) or different variant contaminants

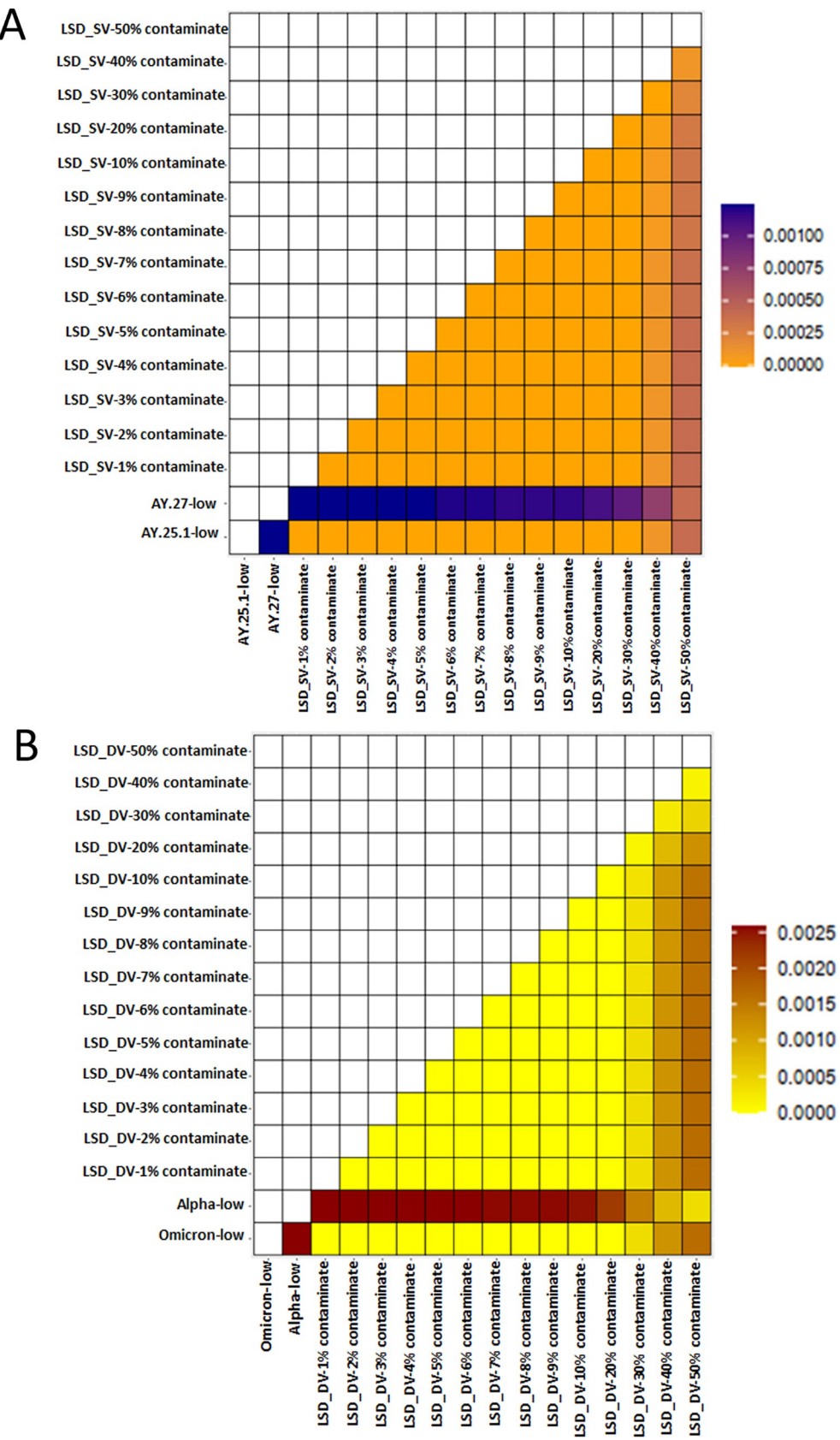

**Fig 2. Global nucleotide comparison of artificially generated contaminated samples and their corresponding clinical SARS-CoV-2 reads as the background samples at a low sequencing depth.** A) A heatmap of the pairwise p-distance comparison of the LSD_SV samples—a delta background sequence (AY.25.1) contaminated with a similar delta contaminant sequence (AY.27). B) A heatmap of the pairwise p-distance comparison of the LSD_DV samples–an omicron background sequence (BA.1) contaminated with an alpha contaminant sequence (B.1.1.7).

(Fig 2B)), differences observed for global nucleotide composition is relatively minor for contamination levels less than 50% (see Fig 2 for the low sequencing depth, S1 and S2 Figs for medium and high sequencing depths respectively).

## The effect of contamination from similar variants on assembly accuracy and lineage calls

The impacts of contamination on SNVs and lineage call outputs for the SARS-CoV-2 genome were assessed by subsampling and mixing the reads from clinical libraries to simulate contaminated genomes at predetermined sequencing depths. Phylogenetic trees were constructed to examine the impact of SNVs found within each subsampled dataset and sequences from the clinical SARS-CoV-2 reads were used as controls. The identified SNVs were plotted with an associated single nucleotide polymorphism (SNP) matrix (Figs 3, S3, and S4). We have highlighted seven quality control metrics (QC metrics) as important metrics in determining contamination thresholds and the effect(s) of sequence contamination on genome completeness and assembly accuracy. These metrics include the number of consensus SNVs obtained from the number of variants in the consensus sequence, the number of consensus 'N', the number of variants SNVs–obtained from the number of gene sequence variations, the number of variants indels, genome completeness, lineage, and Scorpio call, all compared to the reference SARS-CoV-2 strain.

Changes in both the numbers of consensus SNVs and consensus 'N' (number of undetermined data sites) were investigated as essential determinants of assembly accuracy and completeness. For the LSD_SV genomes (12,500 reads), differences in the two aforementioned metrics were observed for the genomes with contamination levels greater than 5% (625 reads) (Fig 3A and Table 2). For instance, the number of consensus SNVs for the clinical delta AY.25.1 relative to the Wuhan reference strain (GenBank accession MN908947.3) was 40, the number of consensus 'N' was 190 and the number of variants SNVs was 45 (Table 2). As the levels of contamination increased, a decrease in the number of SNVs and an increase in the number of consensus 'N's were observed in the contaminated genomes compared to the clinical control samples (Fig 3A and Table 2). Since these are artificially contaminated datasets, we can compare the known source genome sequence. Further, LSD_SV genomes were assigned incorrect lineage calls at contamination levels greater than 30% (3,750 reads). Thus, for LSD_SV genomes, we conclude the contamination threshold for preserving assembly accuracy is 5% while the identified threshold for lineage calls is 30% (Fig 3A and Table 2). See S3 Fig and S1 Table for results obtained for MSD and HSD samples.

## The effect of contamination on assembly accuracy and lineage calls for SARS-CoV-2 sequences for different variants

We investigated the effect of different levels of contamination on SARS-CoV-2 sequences contaminated by different strains (i.e. omicron clinical sample (BA.1), contaminated with an alpha clinical sample (B.1.1.7)). A contamination threshold was identified for changes in SNVs and the number of consensus 'N'—a measure of assembly accuracy and lineage calls at the

A

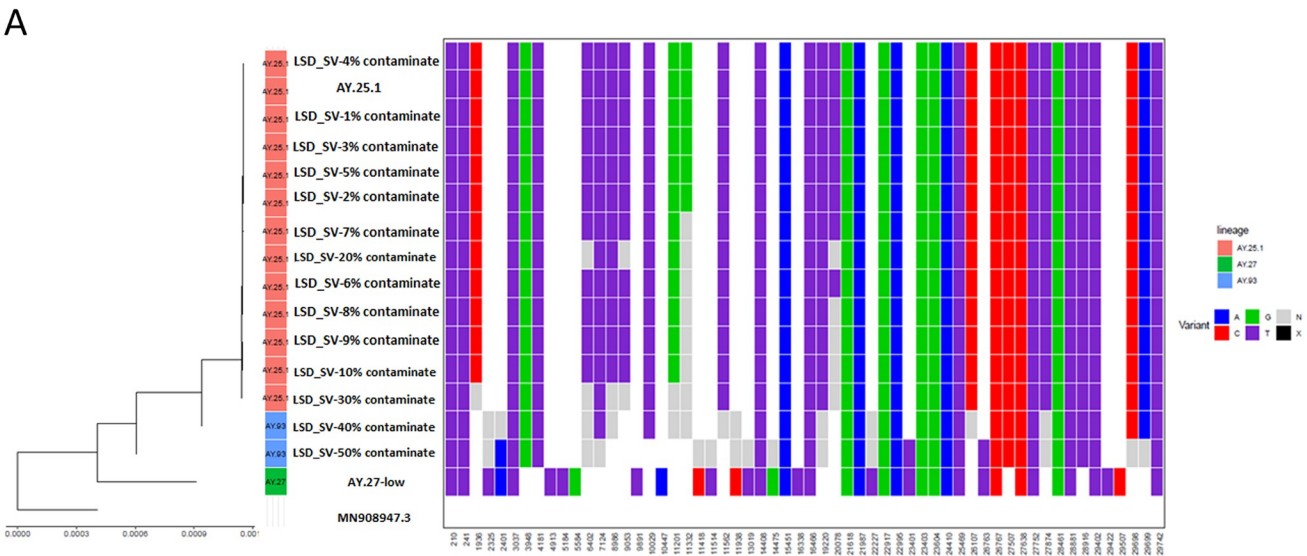

B

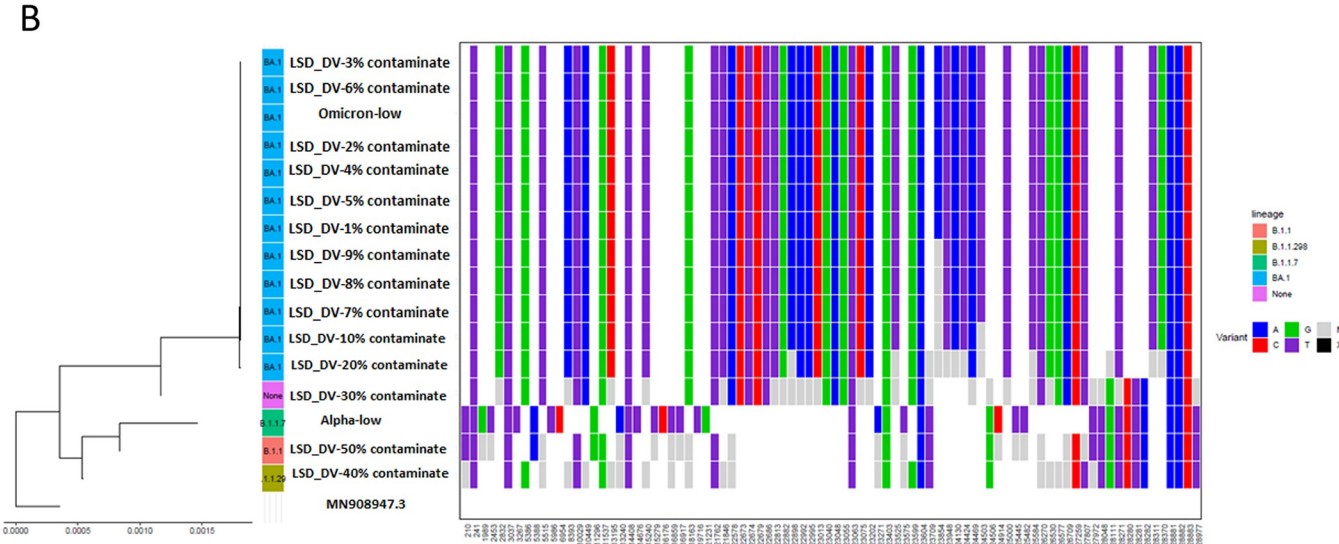

**Fig 3.** Phylogenetic tree and heatmap showing single nucleotide variants (SNVs) at different positions of the SARS-CoV-2 genome for (A) AY.25.1 (delta variant) contaminated with an AY.27 (delta variant) sequence at contamination levels 1–10%, 20%, 30%, 40%, and 50% at low sequencing depth sequence. (B) BA.1 (omicron variant) contaminated with a B.1.1.29 (alpha variant) sequence at contamination levels 1–10%, 20%, 30%, 40%, and 50% at low sequencing depth sequence.

three sequencing depths beyond which the integrity of the data is altered from the source or "true" clinical sample.

At a low sequencing depth (12,500 reads), the number of consensus SNVs for the clinical omicron BA.1 sample relative to the Wuhan reference strain was 56, the number of consensus 'N' was 189, and the number of variants SNVs was 61 (Table 3). Therefore, we investigated the differences in these QC metrics for each of the artificially generated genomes. For the LSD_DV at a contamination level of 7%, it was observed that the number of consensus SNVs changed from 56 to 55, the number of consensus 'N' increased to 190 while the number of variant SNVs decreased at 60 and other QC metrics remained unchanged at this contamination level (Table 3). However, at 30%, the assigned lineage calls for the artificially generated genome

**Table 2. Quality control metrics comparison for artificially subsampled and contaminated genomes of contamination by similar variants at a low sequencing depth–all LSD_SV genomes.**

| Genome | Num of consensus SNVs | Number of consensus 'N' | Number of variants SNVs | Number of variants indel | Mean sequencing depth | Genome completeness | Lineage | Scorpio calls | Watch mutations |
|---|---|---|---|---|---|---|---|---|---|
| AY.25.1_low | 40 | 190 | 45 | 2 | 472.1 | 0.9936 | AY.25.1 | Delta (B.1.617.2-like) | S:G142D,S:L452R |
| LSD_SV-1% contaminate | 40 | 190 | 45 | 2 | 472.1 | 0.9936 | AY.25.1 | Delta (B.1.617.2-like) | S:G142D,S:L452R |
| LSD_SV-2% contaminate | 40 | 190 | 45 | 2 | 472.1 | 0.9936 | AY.25.1 | Delta (B.1.617.2-like) | S:G142D,S:L452R |
| LSD_SV-3% contaminate | 40 | 190 | 45 | 2 | 472.1 | 0.9936 | AY.25.1 | Delta (B.1.617.2-like) | S:G142D,S:L452R |
| LSD_SV-4% contaminate | 40 | 190 | 45 | 2 | 472.1 | 0.9936 | AY.25.1 | Delta (B.1.617.2-like) | S:G142D,S:L452R |
| LSD_SV-5% contaminate | 40 | 190 | 45 | 2 | 472.1 | 0.9936 | AY.25.1 | Delta (B.1.617.2-like) | S:G142D,S:L452R |
| LSD_SV-6% contaminate | 39 | 190 | 44 | 3 | 472.1 | 0.9936 | AY.25.1 | Delta (B.1.617.2-like) | S:G142D,S:L452R |
| LSD_SV-7% contaminate | 39 | 190 | 44 | 3 | 472.1 | 0.9936 | AY.25.1 | Delta (B.1.617.2-like) | S:G142D,S:L452R |
| LSD_SV-8% contaminate | 38 | 191 | 43 | 3 | 472.1 | 0.9936 | AY.25.1 | Delta (B.1.617.2-like) | S:G142D,S:L452R |
| LSD_SV-9% contaminate | 38 | 191 | 43 | 3 | 472.2 | 0.9935 | AY.25.1 | Delta (B.1.617.2-like) | S:G142D,S:L452R |
| LSD_SV-10% contaminate | 38 | 191 | 43 | 3 | 472.1 | 0.9934 | AY.25.1 | Delta (B.1.617.2-like) | S:G142D,S:L452R |
| LSD_SV-20% contaminate | 36 | 194 | 41 | 2 | 472.2 | 0.9935 | AY.25.1 | Delta (B.1.617.2-like) | S:G142D,S:L452R |
| LSD_SV-30% contaminate | 33 | 196 | 38 | 3 | 472.4 | 0.9934 | AY.25.1 | Delta (B.1.617.2-like) | S:G142D,S:L452R |
| LSD_SV-40% contaminate | 29 | 202 | 34 | 2 | 472.3 | 0.9932 | AY.93 | Delta (B.1.617.2-like) | S:G142D,S:L452R |
| LSD_SV-50% contaminate | 28 | 203 | 32 | 2 | 472 | 0.9932 | AY.93 | Delta (B.1.617.2-like) | S:G142D,S:L452R |
| AY.27_low | 39 | 189 | 43 | 3 | 471.7 | 0.9937 | AY.27 | Delta (B.1.617.2-like) | S:G142D,S:L452R |

(LSD_DV) changed from BA.1 to none (Table 3), and this held true for artificial genomes with 40% and 50% contamination. Taken together, for low sequencing depth (LSD_DV), 6% level of contamination (750 reads) was identified as the contamination threshold for the preservation of assembly accuracy while a 20% level of contamination (2,500 reads) was identified as the threshold for accurate lineage call (Fig 3B and Table 3). Results for the MSD and HSD samples can be found in S4 Fig and S2 Table.

To identify the effect of mixture on the artificially subsampled and contaminated genomes, we generated an amino acid mutation heatmap. Mutational profiles and other host-modulating factors have been reported as major contributors to disease severity in COVID-19 [21]. As such, there is a critical need to evaluate the effect of contamination on mutational profiles that may be of clinical importance. The mutational profile compared all defining mutations of the artificially mixed genomes to the clinical SARS-CoV-2 samples and also identified the type/nature of the mutations (conservative in-frame deletion, disruptive in-frame deletion, missense variant, stop gained, and synonymous variant) (Fig 4 and Tables 2 and 3). The amino acid mutation plots reveal the similarity in the mutation profile of each artificially subsampled

**Table 3. Quality control metrics comparison for artificially subsampled and contaminated genomes of contamination by different variants at a low sequencing depth–for all LSD_DV genomes.**

| Genome | Num. of consensus_snvs | Number of consensus 'N' | Number of variants SNVs | Number of variants indel | Mean sequencing depth | Genome completeness | Lineage | Scorpio calls | Watch mutations |
|---|---|---|---|---|---|---|---|---|---|
| BA.1_low | 56 | 189 | 61 | 7 | 470.4 | 0.9937 | BA.1 | Omicron (BA.1-like) | S:del69-70,S:K417N,S:Q493R,S:N501Y,S:P681H,S:P681H11 |
| LSD_DV-1% contaminate | 56 | 189 | 61 | 4 | 470.4 | 0.9937 | BA.1 | Omicron (BA.1-like) | S:del69-70,S:K417N,S:Q493R,S:N501Y,S:P681H,S:P681H |
| LSD_DV-2% contaminate | 56 | 189 | 61 | 4 | 470.4 | 0.9937 | BA.1 | Omicron (BA.1-like) | S:del69-70,S:K417N,S:Q493R,S:N501Y,S:P681H,S:P681H |
| LSD_DV-3% contaminate | 56 | 189 | 61 | 4 | 470.4 | 0.9937 | BA.1 | Omicron (BA.1-like) | S:del69-70,S:K417N,S:Q493R,S:N501Y,S:P681H,S:P681H |
| LSD_DV-4% contaminate | 56 | 189 | 61 | 4 | 470.4 | 0.9937 | BA.1 | Omicron (BA.1-like) | S:del69-70,S:K417N,S:Q493R,S:N501Y,S:P681H,S:P681H |
| LSD_DV-5% contaminate | 56 | 189 | 61 | 4 | 470.4 | 0.9937 | BA.1 | Omicron (BA.1-like) | S:del69-70,S:K417N,S:Q493R,S:N501Y,S:P681H,S:P681H |
| LSD_DV-6% contaminate | 56 | 189 | 61 | 4 | 470.4 | 0.9937 | BA.1 | Omicron (BA.1-like) | S:del69-70,S:K417N,S:Q493R,S:N501Y,S:P681H,S:P681H |
| LSD_DV-7% contaminate | 55 | 190 | 60 | 4 | 470.4 | 0.9936 | BA.1 | Omicron (BA.1-like) | S:del69-70,S:K417N,S:Q493R,S:N501Y,S:P681H,S:P681H |
| LSD_DV-8% contaminate | 55 | 190 | 60 | 4 | 470.4 | 0.9936 | BA.1 | Omicron (BA.1-like) | S:del69-70,S:K417N,S:Q493R,S:N501Y,S:P681H,S:P681H |
| LSD_DV-9% contaminate | 55 | 190 | 60 | 4 | 470.4 | 0.9935 | BA.1 | Omicron (BA.1-like) | S:del69-70,S:K417N,S:Q493R,S:N501Y,S:P681H,S:P681H |
| LSD_DV-10% contaminate | 54 | 191 | 59 | 4 | 470.4 | 0.9936 | BA.1 | Omicron (BA.1-like) | S:del69-70,S:K417N,S:Q493R,S:N501Y,S:P681H,S:P681H |
| LSD_DV-20% contaminate | 46 | 210 | 51 | 4 | 470.6 | 0.993 | BA.1 | Omicron (BA.1-like) | S:del69-70,S:K417N,S:Q493R,S:N501Y,S:P681H,S:P681H |
| LSD_DV-30% contaminate | 34 | 214 | 38 | 4 | 470.7 | 0.9928 | None | Probable Omicron (Unassigned) | S:del69-70,S:Q493R,S:N501Y,S:P681H,S:P681H |
| LSD_DV-40% contaminate | 25 | 214 | 28 | 4 | 470.7 | 0.9928 | B.1.1.298 |  | S:del69-70,S:N501Y,S:P681H,S:P681H,S:T716I,S:S982A |
| LSD_DV-50% contaminate | 26 | 210 | 28 | 4 | 471 | 0.993 | B.1.1 |  | S:del69-70,S:N501Y,S:P681H,S:P681H,S:T716I,S:S982A |
| **B.1.1.7 low** | 40 | 192 | 44 | 4 | 471.1 | 0.9936 | B.1.1.7 | Alpha (B.1.1.7-like) | S:del69-70,S:del144,S:N501Y,S:A570D,S:P681H,S:P681H,S:T716I,S:S982A,S:D1118H |

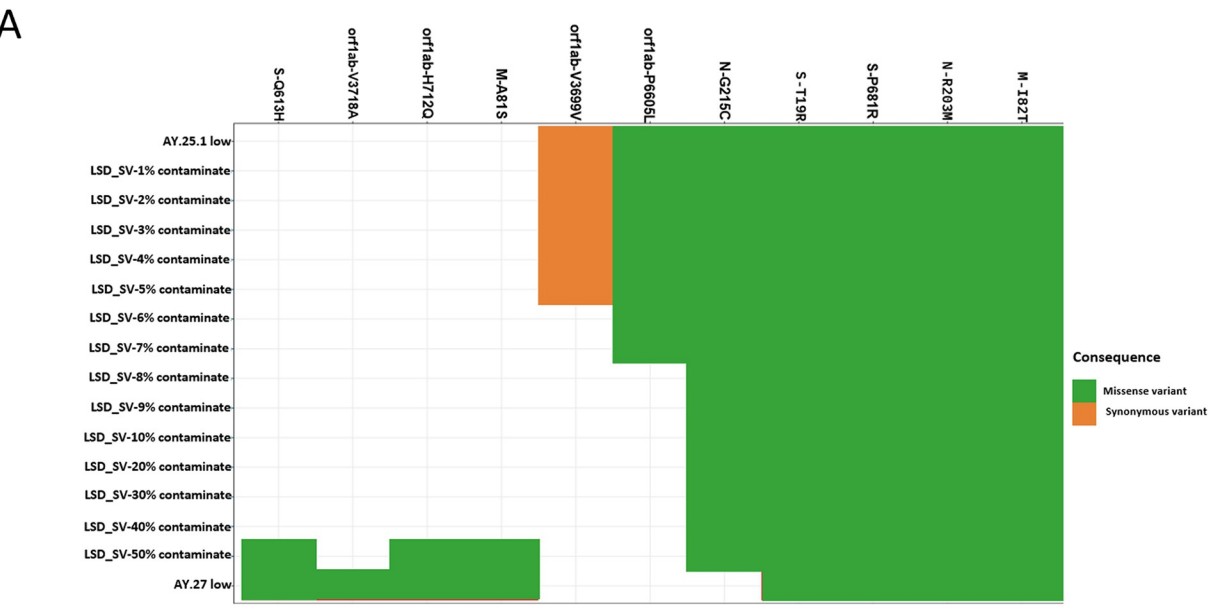

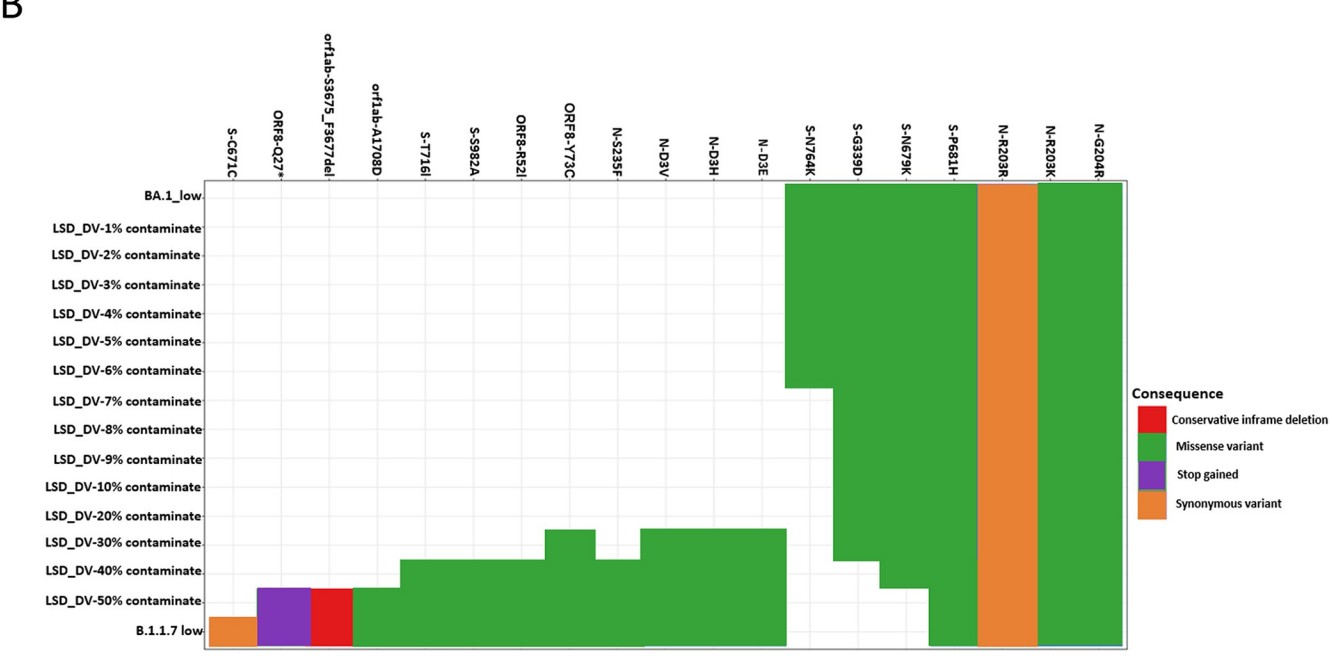

**Fig 4.** Mutational profile comparison of SARS-CoV-2 genome for the clinical genomes to the artificially generated genomes for (A) LSD_SV (AY.25.1 contaminated with an AY.27 variant) sequence at contamination levels 1–10%, 20%, 30%, 40%, and 50%. (B) LSD_DV (BA.1 contaminated with a B.1.1.29 variant) at contamination levels 1–10%, 20%, 30%, 40%, and 50%.

and contaminated genome and the clinical control samples. Samples contaminated by a similar variant (LSD_SV, MSD_SV, and HSD_SV) also had similar mutational profiles while samples with contaminants from different variants (LSD_DV, MSD_DV, and HSD_DV) had different mutational profiles (Tables 1 and 4). It is noteworthy that the artificially subsampled and contaminated genomes mixed from substrains of the same variant had similar mutational

**Table 4. A summary of the identified threshold for the artificially subsampled and contaminated reads as well as the origin of both background and contaminant samples.**

| Standardized term | The identified threshold for assembly accuracy | The identified threshold for lineage call |
|---|---|---|
| LSD_SV | 5 percent | 30 percent |
| MSD_SV | 4 percent | 30 percent |
| HSD_SV | 10 percent | 50 percent |
| LSD_DV | 6 percent | 20 percent |
| MSD_DV | 7 percent | 20 percent |
| HSD_DV | 7 percent | 20 percent |

profiles to the clinical SARS-CoV-2 samples up to and including 5% contamination (Fig 4 and Tables 2 and 4). While the artificially generated subsamples contaminated from a divergent variant had similar mutational profiles to the corresponding clinical SARS-CoV-2 up to and including 6% contamination (Fig 4 and Tables 3 and 4).

In conclusion, for artificial genomes generated by mixing different SARS-CoV-2 variants (e.g., an omicron sample contaminated by an alpha sample), we identified the following thresholds for data integrity: assembly accuracy is maintained at contamination levels up to 6% for LSD (12,500 reads) and 7% for both MSD (25,000 reads) and HSD (50,000 reads) depths. The integrity of lineage calls is more robust, remaining unaltered up to contamination rates of 20% at all sequencing depths.

## Discussion

The scale of sequencing data available in public repositories over the course of the SARS-CoV-2 pandemic is unprecedented. Due to the rapidly evolving nature of the SARS-CoV-2 genome, routine monitoring and public health warnings were crucial in controlling the pandemic. Continuous monitoring and genomic sequencing during the SARS-CoV-2 coronavirus pandemic also hastened the development of the most effective vaccines [22]. However, recurrent mutations in the SARS-CoV-2 genome have tested the efficacy of the vaccines and point to the need for routine updates to both the vaccine targets and vaccination schedules [22,23]. The importance of routine monitoring of SARS-CoV-2 mutations for public health applications cannot be overstated, therefore it is critical that we maintain confidence in the sequences both submitted and pulled from public repositories lest erroneous variants affect major public health decisions [24].

Contaminant-induced mutations have been found and documented in other large-scale genomic studies and it was concluded that these contaminated sequences can spread into and across databases over time [14]. This issue cannot be ignored since genome sequences in public repositories are frequently obtained for comparative studies, forecasting, and decision-making purposes. Therefore, researchers interested in a particular pathogen can collect hundreds of sequences for comparative genomic or phylogenomic investigations in this manner. Lupo et al. demonstrated the presence of mis-affiliated genomes in NCBI RefSeq [25]. While these genomes may not be contaminated in the strictest sense, the dominant organism was not what was expected in the study, leading to problems for downstream analyses and reporting [25]. Despite the findings of these studies, sequences submitted to public repositories/databases are rarely checked for contamination [25]. To further validate the effect of contamination on sequencing data and demonstrate the need for contaminant investigation before data are uploaded to public repositories, this study aimed to identify a contamination threshold for which runs can be considered ideal for upload to public repositories while also offering practical guidelines. While it can be very difficult to identify contamination in a clinical setting, we

completed this study such that the thresholds for contamination identified through the *in silico* experiments can be extrapolated to negative controls in a clinical setting giving the sequence-generating laboratories a guideline to assess before reporting results or uploading data to public repositories.

Since there is no consensus within the scientific community on how to validate assembly accuracy, we investigated the amino acid mutational profile, genome completeness, number of SNVs, number of consensus N, number of variant SNVs, and indels for all samples as a measure of assembly accuracy for this study (Tables 1, 2, S1, and S2)

We further investigated the effect of contamination on the phylogenetic placement and sample relatedness of the artificially subsampled and contaminated genomes (Figs 2, S3, and S4). The results obtained from the phylogenetic analyses are in agreement with the identified contamination thresholds for mutation profile as a measure of assembly accuracy, wherein the artificially subsampled and contaminated genomes with contaminants of less than 5% for LSD_SV and 6% for LSD_DV clustered in the same branch with the corresponding clinical samples. Similar results were also obtained for both MSD_SV and HSD_SV as well as for MSD_DV and HSD_DV. With this observation, we showed that at contamination levels of less than 6%, at all sequencing depths, the artificially subsampled and contaminated genomes were very similar to the genomes generated by the original clinical samples from which they were derived. Thus, we concluded that contamination levels of 5% and below do not affect genome-relatedness and integrity.

By performing a global nucleotide comparison, varying both the levels of simulated contamination and the sequencing depth, we investigated the effect of contamination on the artificially subsampled and contaminated genomes. According to the results obtained from the p-distance pairwise comparison analysis, differences observed for global nucleotide composition among the samples were negligible for contamination levels less than 20% when the metric of interest is simply the lineage assignment irrespective of the sequencing depth and the contamination type. Since p-distance is the proportion of nucleotide sites at which two sequences are different, this result is expected. The analysis performed considers all nucleotides present in the samples compared without any regard for the origin of the nucleotide (i.e., background or contaminant). However, it is noteworthy that with contamination levels greater than 20%, differences were observed at the global nucleotide levels when compared to the original clinical samples at all sequencing depths for both types of contaminants (Figs 2, S1, and S2).

Studies have identified the importance of lineage tracking and its role in providing answers to evolutionary questions about the SARS-CoV-2 genome [26,27]. The extensive recombination between SARS-CoV-2 strains, first identified by so-called "deltacron" lineages with diagnostic mutations associated with both the delta and omicron variants have become identified with increasing frequency since late 2021, and the emergence of the omicron variant [28,29]. Thus, the accurate assignment of lineage calls for SARS-CoV-2 lineages is important, as these lineages also offer insights for clinicians and public health personnel during an outbreak of infection. Based on the above notion, we investigated the effect that the different levels and types of contamination had on the accuracy of lineage calls (Tables 1, 2, S1, and S2). Our results showed that regardless of the type of contaminant (similar or different sequences), a 20% contamination threshold was the maximum amount permissible for accurate lineage calls (Tables 1, 2, S1, and S2).

Foreign sequences can be introduced at many different stages of the sequencing process, from organism culture to data processing [14]. Here, we offer some practical guidelines on how to track contaminants during sequencing experiments and offer recommendations to researchers on steps to take before uploading reads to public repositories. We recommend that researchers include a negative control in the following steps: (i) nucleic acid extraction, (ii)

nucleic acid amplification (if applicable), and (iii) library preparation steps. By having multiple negative controls introduced at different stages of the sequencing experiment, the source of contamination may be identified. We also recommend that these negative controls be carried forward to the data processing steps so that if contamination occurs, the amount of sequenced data present in the negative controls could be investigated and used to determine the appropriate contamination threshold based on the objective(s) of the sequencing experiment in question. Upon the investigation of the negative controls, it is advised that the number of reads mapped in the negative controls should be checked to ensure that the percentage of reads mapped do not exceed the recommended threshold (determined by the average read depth of samples in the run). This practice will also be useful to determine if samples were contaminated by multiple strains.

As this study is the first of its kind, we are aware that these identified thresholds may change as more sequence data become available and as more studies expand on and investigate the parameters required for assembly accuracy and lineage calls. However, we hope that having a standardized method for determining the integrity of genomes and lineage calls will provide a benchmark below which imperfect runs may be considered robust for reporting results to both stakeholders and public repositories, thereby reducing the need for repeat or wasted runs. In this study, we investigated contamination thresholds for SARS-CoV-2 samples generated by Nanopore sequencing by conducting *in silico* analyses. We identified a contamination threshold of 5% that does not compromise the integrity of the genome and a contamination threshold of 20% for lineage calls. Our results suggest the establishment of a stricter threshold if the preservation of assembly accuracy is of utmost importance. Future larger-scale studies are warranted to systematically investigate the effects of contamination on both SARS-CoV-2 reads and other viral and bacterial sequences to serve as a check step for sequencing upload.

## Supporting information

**S1 Fig. Global nucleotide comparison of artificially generated contaminated samples and their corresponding clinical SARS-CoV-2 reads as the background samples at a medium sequencing depth.** A) A heatmap of the pairwise p-distance comparison of the MSD_SV samples—a delta background sequence (AY.25.1) contaminated with a similar delta contaminant sequence (AY.27). B) A heatmap of the pairwise p-distance comparison of the MSD_DV samples–an omicron background sequence (BA.1) contaminated with an alpha contaminant sequence (B.1.1.7).
(TIF)

**S2 Fig. Global nucleotide comparison of artificially generated contaminated samples and their corresponding clinical SARS-CoV-2 reads as the background samples at a high sequencing depth.** A) A heatmap of the pairwise p-distance comparison of the HSD_SV samples—a delta background sequence (AY.25.1) contaminated with a similar delta contaminant sequence (AY.27). B) A heatmap of the pairwise p-distance comparison of the HSD_DV samples–an omicron background sequence (BA.1) contaminated with an alpha contaminant sequence (B.1.1.7).
(TIF)

**S3 Fig. Phylogenetic tree and heatmaps showing single nucleotide variation at different positions of the SARS-CoV-2 genome for** (A) a delta variant (AY.25.1) contaminated with another delta variant (AY.27) sequence at contamination levels 1–10%, 20%, 30%, 40%, and 50% for medium sequencing depth and (B) an omicron variant (BA1) contaminated with an alpha contaminant sequence (B.1.1.7) at contamination levels 1–10%, 20%, 30%, 40%, and

50% for medium sequencing depth (25,000 reads).
(TIF)

**S4 Fig.** Phylogenetic tree and heatmaps showing single nucleotide variation at different positions of the SARS-CoV-2 genome for (A) a delta variant (AY.25.1) contaminated with another delta variant (AY.27) sequence at contamination levels 1–10%, 20%, 30%, 40%, and 50% for high sequencing depth and (B) an omicron variant (BA1) contaminated with an alpha contaminant sequence (B.1.1.7) at contamination levels 1–10%, 20%, 30%, 40%, and 50% for high sequencing depth (50,000 reads).
(TIF)

**S5 Fig.** Mutational profile comparison of SARS-CoV-2 genome for the clinical genomes to the artificially generated genomes for (A) MSD_SV and (B) (AY.25.1 contaminated with an AY.27 variant) sequence at contamination levels 1–10%, 20%, 30%, 40%, and 50%. (C) MSD_DV and (D) HSD_DV (BA.1 contaminated with a B.1.1.29 variant) at contamination levels 1–10%, 20%, 30%, 40%, and 50%.
(TIF)

**S1 Table. A.** Quality control metrics comparison for artificially subsampled and contaminated genomes of contamination by similar variants at a medium sequencing depth–for all MSD_SV genomes. **B.** Quality control metrics comparison for artificially subsampled and contaminated genomes of contamination by different variants at a medium sequencing depth–for all MSD_DV genomes.
(DOCX)

**S2 Table. A.** Quality control metrics comparison for artificially subsampled and contaminated genomes of contamination by similar variants at a high sequencing depth–for all HSD_SV genomes. **B.** Quality control metrics comparison for artificially subsampled and contaminated genomes of contamination by different variants at a high sequencing depth–for all HSD_DV genomes.
(DOCX)

## Acknowledgments

We owe a debt of gratitude to Dr. Anna Majer and the DNA core team at the National Microbiology Laboratory for sequencing the clinical samples utilized in this study. We sincerely thank Dr. Andrea Tyler for her contributions to the experimental design of this study and we appreciate the insightful discussions had with Dr. David Alexander and Dr. Kerry Dust of Cadham Provincial Laboratory.

## Author Contributions

**Conceptualization:** Ayooluwa J. Bolaji, Ana T. Duggan.

**Data curation:** Ayooluwa J. Bolaji.

**Formal analysis:** Ayooluwa J. Bolaji.

**Investigation:** Ayooluwa J. Bolaji.

**Methodology:** Ayooluwa J. Bolaji, Ana T. Duggan.

**Project administration:** Ayooluwa J. Bolaji.

**Resources:** Ana T. Duggan.

**Supervision:** Ayooluwa J. Bolaji, Ana T. Duggan.

**Validation:** Ayooluwa J. Bolaji.

**Visualization:** Ayooluwa J. Bolaji.

**Writing – original draft:** Ayooluwa J. Bolaji.

**Writing – review & editing:** Ayooluwa J. Bolaji, Ana T. Duggan.

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
