## [Decision Letter · Decision Letter 0]

22 Nov 2023

Dear Duggan,

Thank you very much for submitting your manuscript "In silico analyses identify sequence contamination thresholds for Nanopore-generated SARS-CoV-2 sequences" for consideration at PLOS Computational Biology.

As with all papers reviewed by the journal, your manuscript was reviewed by members of the editorial board and by several independent reviewers. In light of the reviews (below this email), we would like to invite the resubmission of a significantly-revised version that takes into account the reviewers' comments.

We cannot make any decision about publication until we have seen the revised manuscript and your response to the reviewers' comments. Your revised manuscript is also likely to be sent to reviewers for further evaluation.

Sincerely,

William Stafford Noble

Section Editor

PLOS Computational Biology

William Noble

Section Editor

PLOS Computational Biology

The points made by Reviewer 3 at the end of the review comments (those to the editors) should be resolved by this study. We quote them as follows: The sequence of each virus is different as differed in human fingerprints; now the question is whether these differences should be treated as contamination, mutation, new strain or new reference.

Reviewer's Responses to Questions

**Comments to the Authors:**

Reviewer #1: In this work, Bolaji and Duggan simulate contaminated Nanopore datasets of SARS-CoV2 genomes at different contamination levels and sequencing depths to study the effect of sample contamination on genome integrity and lineage calls. The raw reads are real but were mixed up at the FASTQ file step. After assembly with the ARTIC pipeline (following the Nanopolish variant calling path), those were further processed with ncov-tools and Pangolin. All datasets have been made available by the authors on Zenodo.

Overall, the study was well designed (even if modest in ambition) and the reported interpretations are in line with the results. I have some concerns regarding the significance of the conclusions, as well as the length and clarity of the manuscript, but I did not identify any major flaw in the authors' work that would make the study unpublishable after revision.

* Scientific significance

1. The objective of the authors is to provide contamination level thresholds under which genome integrity and lineage calls would be unaffected (e.g., lines 414-415). Moreover, they study the matter at different sequencing depths, from one typical multiplexed experiment (50k reads per sample) down to 1/4 of such an experiment (12.5k reads). I see the point but, in real life, one does not know the level of contamination. Therefore it would be more useful to identify clues in raw read datasets and/or assembled genomes that would help detecting and pinpointing sample contamination in a real setup (lines 456-457). In that respect the authors' suggestion about negative controls makes sense and could be further developed (lines 447-451).

2. On a related note, the authors rightfully raise the issue of recombinant genomes, for example "deltacron" viruses (lines 430-434). It is true that, taken at face value, contaminated samples could yield pseudo-recombinant genomes in terms of hybrid combinations of variant-specific SNVs (and thus intermediate p-distances with background "pure" samples). Yet, the mutational profile along the consensus genome should be different between true recombinants and contaminated assemblies. Indeed (maybe naively speaking), I would expect recent recombinant genomes to be composed of two different halves, each one corresponding to a "donor" variant whereas contaminated assemblies would display a more homogeneous mixture of diagnostic nucleotides and a stronger propension to have ambiguous nucleotides ('N') at these variable positions. Maybe these points would gain to be discussed in the manuscript.

3. While the experimental design is sound, it is also modest, in the sense that only four starting variants were considered as the basis for the whole study (Table 1 and lines 452-454). What would be the effects of using different variants? Can the issue be really reduced to "similar variants" vs "different variants" without consideration for other p-distances between variants? Owing to these issues, it is difficult to determine whether the reported thresholds are generalizable to other (real) situations. In the same line of thought (hence in the same comment), there is no statistical framework at all in the study. This leads to sentences that are probably devoid of an actual statistical significance, e.g., "a decrease in the number of SNVs (from 40 to 39) and an increase in the number of consensus ‘n’ (from 189 to 190) were observed" (lines 236-237). I think that these claims should either be avoided or be backed-up by statistics.

* Manuscript organization and length

4. The results are faithfully reported but the study structure makes its reporting quite repetitive. There are similar blocks of text (and figures and tables) for each combination of sequencing depth (LSD, MSD, HSD) and p-distance (SV, DV) and even repetitions within a single combination, e.g., "In conclusion, for contamination by a similar SARS-CoV-2 variant, the contamination threshold identified for lineage call was 30% for both LSD_SV and MSD_SV and 50% for HSD_SV genomes. However, for genome integrity, the contamination threshold was 5% for low, 4% for medium, and 10% for high sequencing depths (Figures 4 A & B)" (lines 242-246). My opinion is that the net amount of knowledge brought by the study does not warrant such a long and repetitive manuscript. While it would totally make sense for a student's thesis, for a published article, it is inadequate. Hence, I would suggest compressing the manuscript by only pointing to the salient differences with respect to a reference combination.

5. Related to this, the groupings in Supplementary figures are different from those in main figures, i.e., grouped by variant p-distance instead of sequencing depth. This is of course acceptable but quite confusing for the reader. On the same point, legend of Figure 2 is likely incorrect: panel B should read LSD and not HSD. Finally, I would suggest using the authors' abbreviations here too: LSD_SV (instead of LSD alone) and LSD_DV (instead of HSD).

6. Back to manuscript length, I am doubtful about the usefulness of the first part of the results ("Global nucleotide comparison at different levels of contamination for different sequencing depths") and the associated heat maps. Which kind of insights and/or thresholds can be derived from this? Basically, these p-distances are already summarized in the phylogenetic trees, which are much more interesting as they further show the SNVs. That is why I would advise the authors to remove or considerably shorten this section.

7. The whole section of the discussion about mutational profiles (lines 385-404) should be transferred to the results because this where such a paragraph naturally belongs. This reorganization would also fix the lack of a proper introduction for Figure 4.

8. Regarding Figure 5, it is quite large for a rather modest gain of insight. This is partly due to the choice of using pie-charts, which are known to have a very bad ratio of information content to chart size.

* Manuscript clarity

All the remaining comments are minor but should be addressed to improve the clarity of the manuscript. I list them below in line order.

- lines 95-97: "NGS runs are rarely repeated for reasons including limited funds to repeat expensive library preparation reactions and NGS remains relatively expensive, even when samples are multiplexed." Seemingly internally redundant sentence. Please rephrase.

- line 101: "the results of genomic analyses of organisms" I agree with the idea, but if this sentence concerns viruses, these are not organisms.

- lines 119 and following: It is difficult to understand where the empirical reads that are mixed up to generate artificial datasets come from. Are these from only four discrete samples among the 60+752 samples sequenced in the study? Please clarify this part of the design because it is confusing. Lines 172-173 are also related to this issue and should be clarified.

- lines 126, 132, 175: "read length(s)" I do not see the point of mentioning read length in this context and I am pretty sure that at least at line 132, "length" should read "depth". Please check.

- line 128: "Random subsampled artificial sequences" Ambiguous wording that does not convey the idea of artificial FASTQ files mixing (subsampled) raw reads from different empirical samples. Please rephrase and do not use "sequences" for "reads" (also on line 348). On a related note, Figure 1 is not that clear because the idea of mixing reads is not explicitly represented.

- line 130: "15 different contamination levels" Correct but including no contamination at all, i.e., with background reads only. Please amend.

- line 152-153: "Reads were mapped to the reference 153 SARS-CoV-2 genome NCBI GenBank accession (MN908947)" For non-experts of SARS-CoV2 (and viruses in general), please explain why this step is needed and its role in calling the different types of SNVs discussed later (consensus SNVs vs variant SNVs).

- line 159 (and elsewhere): "clinical samples" is ambiguous (see my other comments about details of the experimental design). Please clarify. Also, was there any ambiguity in these genuine "pure" samples? I think so if I look at Tables (e.g., 190 'N' for AY.25.1_low in Table 1). How so?

- lines 161-162: "The p-distance option was chosen as input for the Model/Method setting while the default options were chosen for the other settings." Ok, but is it an observed distance or a model-corrected distance? Please specify.

- lines 185-186: "not substantial for contamination levels less than 20%" To me, it seems difficult to provide a single number for all the combinations (rather 20 to 40% depending on the case). Please check.

- line 204 (legend of Figure 2): It would be useful to provide the p-distances between pairs of background strains in full numbers (not only in color shade).

- lines 211-212: "by creating in silico artificial mixtures of reads to simulate contaminated genomes" I guess these mixtures were the same as before (section about Global comparison). If so, please rephrase because these look newly generated considering the introductory sentence.

- lines 221-222: For non-SARS experts, please explain the difference between consensus SNVs and variant SNVs (see my other comment on the issue above). I would have imagined that these SNVs are called against their respective background strain, but this does not look to be the case here.

- line 225: "for all samples" appears useless and confusing. Please delete.

- lines 230-332: A verb is apparently missing in the sentence, e.g., "were observed". Please check.

- line 247 (and other table legends): Most columns are not explained in the text and some are nor even used (e.g., lineage note, Scorpio call). Please check.

- line 306: "low sequencing depth sequence" should apply to both panels A and B, not only B. Please check.

- lines 321-324 (Figure 4): I fail to understand why some "mutations" are found across all strains and artificial samples. What was the reference genome?

- line 350: "for different sequences" I would have written "for both similar and different variants". Please check.

- lines 366-367: Please clarify the sentence because "lest" is confusing here. If it means "unless", it does not make sense in this context.

- lines 382-383 (and elsewhere): "genome integrity" I am not sure about this wording, which hints at some biological phenomenon. In my opinion, it is more a matter of "assembly accuracy" in this case.

- line 386: "the clinical samples and the artificially subsampled genomes" To me, the main variable is not the subsampling (because clinical samples are also subsampled) but rather the idea of artificial mixture". Please amend.

- line 444: "linage" => lineage

Reviewer #2: In silico analyses identify sequence contamination thresholds for Nanopore-generated SARS-CoV-2 sequences

Synopsis:

The authors attempted to benchmark a method for determining the integrity of genomes and lineage calls of SARS-CoV-2 using ONT sequencing under contamination conditions of different viral strains. Systematic investigation of the effects of contamination on SARS-CoV-2 sequencing reads is sound, with the following recommendations:

Line 120: “Amplicons generated using tiling PCR”. Please explain why using amplicons. What primers were used?

Line 123: “60 SARS-CoV-2 samples sequenced on a MinION device and 752 samples sequenced on a GridION device”. Do they represent known Omicron, Delta and Alpha sequences used in the study?

Line 130: “15 different contamination”. I counted 14 levels?

Line 132: “three read lengths”. Do you mean depths?

Line 154: “Pangolin (version 4.0.3, pangoLEARN) (version 1.2.333)”. Need a reference to cite authors.

Line 155: “consensus sequences”. What criteria to call a consensus sequence? It forms an important point in your results.

Line 158: No phylogenetic analyses explained.

Line 172: Can you tell the mean read length too?

Line 173: “768 SARS-CoV-2 clinical samples”. It doesn’t add up for 60+752 samples in Line 123. Please explain.

Line 222: the number of consensus ‘n’. Please briefly explain how to call N in Methods, due to its importance in results. Is it 100% missing bp across all reads?

Line 345: Do you mislabel in Figure 5 between light and dark blue colors?

Line 440: From “linage calls” to lineage calls.

Methods: Should also explain Scorpio call.

Discussion: Please avoid repeating results and figures/tables in this section. Some might be better in the Results section.

Discussion: The authors used consensus sequences, instead of de novo assembly. In my understanding, it is difficult to differentiate between contaminant and recombinant viruses. [Recombinant virus, please see “World Health Organization. 2022. TAG-VE statement on Omicron sublineages BQ.1 and XBB, 27 October 2022. https://www.who.int/news/item/27-10-2022-tag-ve-statement-on-omicron-sublineages-bq.1-and-xbb.”] Although ONT long reads can define recombinants between viral strains using de novo assembly, this study doesn’t consider this to rule out recombination of a disease-causing virus from contaminant viruses. Please discuss this and provide future improvement.

If applicable, it would be nice to confirm truncated variants in Figure 4 and Figure S5 by other sequencing, especially illumina sequencing to see if the ONT error rates and corresponding pipeline can impact those variants between clinical and contaminant sequences.

Should it be better to develop workflow to identify long reads with variants each representing different strains, instead of a consensus sequence of a single strain or artefactual hybrid strain? Should ONT reads need to be validated by using sanger sequencing due to its ONT bp accuracy?

Discuss how to rule out multiple strain infection in a sample from a contamination event. Can it be co-infected with different strains within a sample?

Reviewer #3: NONE

**Have the authors made all data and (if applicable) computational code underlying the findings in their manuscript fully available?**

Reviewer #1: Yes

Reviewer #2: Yes

Reviewer #3: Yes

PLOS authors have the option to publish the peer review history of their article (what does this mean?). If published, this will include your full peer review and any attached files.

Reviewer #1: **Yes: **Denis BAURAIN

Reviewer #2: No

Reviewer #3: No
---

## [Decision Letter · Decision Letter 1]

3 Apr 2024

Dear Duggan,

Thank you very much for submitting your manuscript "In silico analyses identify sequence contamination thresholds for Nanopore-generated SARS-CoV-2 sequences" for consideration at PLOS Computational Biology. As with all papers reviewed by the journal, your manuscript was reviewed by members of the editorial board and by several independent reviewers. The reviewers appreciated the attention to an important topic. Based on the reviews, we are likely to accept this manuscript for publication, providing that you modify the manuscript according to the review recommendations.

Sincerely,

Jinyan Li

Academic Editor

PLOS Computational Biology

William Noble

Section Editor

PLOS Computational Biology

Reviewer's Responses to Questions

**Comments to the Authors:**

Reviewer #1: I have finally found some time to take a closer look at the revision. I apologize for the delay and I thank the authors for the changes they have made to address my previous comments. I think the revised manuscript is a huge step in the good direction but I still see a number of issues (some already raised in my original review) that should be taken care of before acceptance.

* "(not so) Minor issues"

1. Statistics are inexistant and this is annoying when trying to define thresholds. If I understand correctly, the authors have selected a number of metrics and the thresholds they eventually propose are the highest contamination proportions that do not affect any of these metrics in any analysis (i.e., across the two kinds of strain mixture and three sequencing depths). It makes sense as it is quite conservative, but still very empirical. Given the experimental design, I do not have a specific suggestion to improve this aspect, but I think I must underline it. For example for LSD_SV genomes around 5% contamination [lines 209-211 vs Table 2], most metrics are quite robust.

2. I fail to understand how negative controls would work in practice. If a blank sample is contaminated by an non-targeted viral variant, the latter will occupy 100% of the sequencing reads. How to apply the identified thresholds then? Moreover, blanks can only control for one case of contamination (lab-introduced foreign sequences). Such an approach cannot control for samples contaminated at the time of collection (or at least before sample processing in the lab). Could the authors better explain the way they would use their thresholds to put away samples considered as too contaminated for deposit in public databases?

3. Some comments in the text do not seem to be compatible with the methods supposed to have been used. For example, about the global comparison and the heat map [lines 370-371], "The analysis performed considers all nucleotides present in the samples compared without any regard for the origin of the nucleotide (i.e., contaminant or not)." I am not sure about that: I understood that the authors compared the whole genomes assembled from the contaminated reads: [lines 159-160] "Aligned nucleotide consensus genome sequences of both the clinical samples and the artificially generated genomes were imported to MEGA11 software to calculate pairwise distance." With (only) 20% of contaminant reads, one can expect that the consensus assembled genome will nearly be the correct one, as indeed shown in the trees.

4. In Table 4, HSD_SV is correctly called for its lineage up to 50% contamination. I agree that it is what the results show in this case, but this does not make much sense. When two strains are mixed-up in equal amounts of reads, what should be the expected result? Here the lineage calling algorithm provides a result that apparently matches "the true strain" but there is no hard reason for this. It must be very dependant on the specific polymorphisms shared (or not) by the two closely related strains considered in the mixture.

* Clarity issues

Figure 1 is useful to understand the experimental design but the mixing of viral strains could be better represented because there actually are only two combinations, not four, as possibly suggested by the figure.

[lines 232-235] "(...) the number of consensus SNVs for the clinical omicron BA.1 sample was 56, the number of consensus ‘N’ as a measure of missing nucleotide was 189 and the number of variants SNVs was 61 (Table 3) when compared against the reference Wuhan strain (RefSeq NC_045512.2)." These results are more detailed than in the previous section, which does not make too much sense as they come second. At the opposite [lines 269-270] "assembly accuracy is maintained at contamination levels up to 6% for LSD (12,500 reads) and 7% for both MSD (25,000 reads) and HSD (50,000 reads) depths" has not been mentioned before this small conclusion, except as [lines 218-219] "See supplementary Figure 3 and Supplementary Table 1 for results obtained for MSD and HSD samples at different sequencing depths." Please try to balance the level of details throughout the text, i.e., avoid repeating the same result thrice whereas another important bit is only mentioned in a small conclusion.

[lines 247-248] "To identify the effect that artificial mixture had on both the clinical SARS-CoV-2 genomes and the artificially subsampled genomes" Why the "both"? When were the clinical genomes mixed-up in the experiments?

[lines 261-263] "(...) the artificially subsampled genomes that contained less than 6% of contaminant from a substrain of the same variant had similar mutational profiles to the clinical SARS-CoV-2 samples at all levels of contaminations." I don't understand the sentence: at <6% or at all levels?

[lines 264-266] "While the artificially generated subsampled genomes that contained less than 7% of contaminant from a divergent variant had similar mutational profiles to the corresponding clinical SARS266 CoV-2 samples at all levels of contamination." Same issue and no verb in the sentence.

[line 353 and elsewhere] "artificially subsampled genomes" The authors use this wording in several places but, in my opinion, the main point is that these sample include contaminant reads. Thus, it would better read as "artificially subsampled and contaminated genomes". Similarly [line 194], "by subsampling the reads" should rather be "by subsampling and mixing the reads". Finally, in the same area of the text, as explained in my original review, these sentences are misleading because the discussed samples are in principle the same as those used in the previous section (about global comparison). Thus, their generation should have been explained before this section and not detailed here.

The legend for Figure S5 is missing. Moreover, please check the uniformity of the legends for all figures belonging to the same type of analysis (i.e., there exist three to six variants of each figure).

The following phrases should be clarified too:

- [line 24] "maintain genomic sequence integrity"

- [line 172] "The distance (proportion) of nucleotide sites"

- [line 328] "lest" [already mentioned in my original review]

- [line 556] "similar variants" [while the used code is HSD_DV]

* Other details

[lines 208-209] "consensus ‘N’ (number of missing data sites)" Indeed, but I would rather say "undetermined" than "missing" because these sites are too polymorphic to be determined in the consensus genome due to contamination.

[line 128] "It is of note that in this study, the number of reads was used as a proxy for sequencing depth." This sentence can be safely deleted, since this precision is already mentioned a bit above [lines 99-101].

The following precisions could also be deleted:

- [line 23] "Through a series of analyses"

- [line 219] "at different sequencing depths"

- [lines 245-246] "at different sequencing depths"

- [line 255] "We believe that"

- [lines 287-288] "low sequencing depth sequence (12,500 reads)"

- Table 4 [head of column 2] "call" [it is incorrect]

The following typos should be fixed:

- [line 125] delete extra closing parenthesis

- [line 127] contaminate -> contaminating [or contaminant]

- [line 160] distance -> distances

- [line 176] composition negligible -> composition is negligible

- [line 199] served -> used

- [line 206] indel -> indels

- [line 207] call all -> call, all

- [line 212] levels -> level

- [lines 228 and 305] add missing closing parenthesis

- [line 401] practise -> practice

Reviewer #2: Line 115, add this phrase at the beginning, “Due to low quantities of viral genomic materials in clinical specimens,”

Line 348, fix font size.

-END-

**Have the authors made all data and (if applicable) computational code underlying the findings in their manuscript fully available?**

Reviewer #1: Yes

Reviewer #2: Yes

PLOS authors have the option to publish the peer review history of their article (what does this mean?). If published, this will include your full peer review and any attached files.

Reviewer #1: **Yes: **Denis BAURAIN

Reviewer #2: No

Figure Files:

Data Requirements:

Reproducibility:

References:

---

## [Decision Letter · Decision Letter 2]

14 Jul 2024

Dear Duggan,

We are pleased to inform you that your manuscript 'In silico analyses identify sequence contamination thresholds for Nanopore-generated SARS-CoV-2 sequences' has been provisionally accepted for publication in PLOS Computational Biology.

Best regards,

Jinyan Li

Academic Editor

PLOS Computational Biology

William Noble

Section Editor

PLOS Computational Biology

Reviewer's Responses to Questions

**Comments to the Authors:**

Reviewer #1: Thank you for the answers to my comments.

I can now accept your manuscript.

However, here are three remaining points that you might want to fix in the final version:

- line 99: bins -> amounts, quantities (or even depths)?

- line 321 (Table 4): useful but the numbers pertaining to MSD_SV and HSD_SV only appear here, not above, in contrast to those of the four other combinations, which are properly introduced.

- in all sections about negative controls (lines 103-105, 352-354, 406-409, 418-420), I still fail to see how meaningful percentages (to be compared with the thresholds determined here) will be extracted from blanks contaminated by one undesired strain (see my comment #2 in the previous round of review). Indeed, as this contaminant would not be mixed up with another, true strain, it would occupy 100% of the sequencing reads.

**Have the authors made all data and (if applicable) computational code underlying the findings in their manuscript fully available?**

Reviewer #1: Yes

PLOS authors have the option to publish the peer review history of their article (what does this mean?). If published, this will include your full peer review and any attached files.

Reviewer #1: **Yes: **Denis BAURAIN

---

## [Editor Report · Acceptance letter]

13 Aug 2024

PCOMPBIOL-D-23-01522R2 

In silico analyses identify sequence contamination thresholds for Nanopore-generated SARS-CoV-2 sequences

Dear Dr Duggan,

I am pleased to inform you that your manuscript has been formally accepted for publication in PLOS Computational Biology. Your manuscript is now with our production department and you will be notified of the publication date in due course.

With kind regards,

Anita Estes
